# Strength Training versus Stretching for Improving Range of Motion: A Systematic Review and Meta-Analysis

**DOI:** 10.3390/healthcare9040427

**Published:** 2021-04-07

**Authors:** José Afonso, Rodrigo Ramirez-Campillo, João Moscão, Tiago Rocha, Rodrigo Zacca, Alexandre Martins, André A. Milheiro, João Ferreira, Hugo Sarmento, Filipe Manuel Clemente

**Affiliations:** 1Centre for Research, Education, Innovation and Intervention in Sport (CIFI2D), Faculty of Sport of the University of Porto, Rua Dr. Plácido Costa, 91, 4200-450 Porto, Portugal; jneves@fade.up.pt (J.A.); rzacca@fade.up.pt (R.Z.); up201900293@fade.up.pt (A.M.); up201507435@fade.up.pt (A.A.M.); 2Department of Physical Activity Sciences, Universidad de Los Lagos, Lord Cochrane 1046, Osorno 5290000, Chile; r.ramirez@ulagos.cl; 3Centro de Investigación en Fisiología del Ejercicio, Facultad de Ciencias, Universidad Mayor, San Pio X, 2422, Providencia, Santiago 7500000, Chile; 4REP Exercise Institute, Rua Manuel Francisco 75-A 2 °C, 2645-558 Alcabideche, Portugal; contacto@repinstitute.com; 5Polytechnic of Leiria, Rua General Norton de Matos, Apartado 4133, 2411-901 Leiria, Portugal; tiago.rocha@ipleiria.pt; 6Porto Biomechanics Laboratory (LABIOMEP-UP), University of Porto, Rua Dr. Plácido Costa, 91, 4200-450 Porto, Portugal; 7Coordination for the Improvement of Higher Educational Personnel Foundation (CAPES), Ministry of Education of Brazil, Brasília 70040-020, Brazil; 8Superior Institute of Engineering of Porto, Polytechnic Institute of Porto, Rua Dr. António Bernardino de Almeida, 431, 4249-015 Porto, Portugal; 1200638@isep.ipp.pt; 9Faculty of Sport Sciences and Physical Education, University of Coimbra, 3040-256 Coimbra, Portugal; hugo.sarmento@uc.pt; 10Escola Superior Desporto e Lazer, Instituto Politécnico de Viana do Castelo, Rua Escola Industrial e Comercial de Nun’Álvares, 4900-347 Viana do Castelo, Portugal; 11Instituto de Telecomunicações, Department of Covilhã, 1049-001 Lisboa, Portugal

**Keywords:** flexibility, mobility, joints, resistance training, plyometrics

## Abstract

(1) Background: Stretching is known to improve range of motion (ROM), and evidence has suggested that strength training (ST) is effective too. However, it is unclear whether its efficacy is comparable to stretching. The goal was to systematically review and meta-analyze randomized controlled trials (RCTs) assessing the effects of ST and stretching on ROM (INPLASY 10.37766/inplasy2020.9.0098). (2) Methods: Cochrane Library, EBSCO, PubMed, Scielo, Scopus, and Web of Science were consulted in October 2020 and updated in March 2021, followed by search within reference lists and expert suggestions (no constraints on language or year). Eligibility criteria: (P) Humans of any condition; (I) ST interventions; (C) stretching (O) ROM; (S) supervised RCTs. (3) Results: Eleven articles (*n* = 452 participants) were included. Pooled data showed no differences between ST and stretching on ROM (ES = −0.22; 95% CI = −0.55 to 0.12; *p* = 0.206). Sub-group analyses based on risk of bias, active vs. passive ROM, and movement-per-joint analyses showed no between-protocol differences in ROM gains. (4) Conclusions: ST and stretching were not different in their effects on ROM, but the studies were highly heterogeneous in terms of design, protocols and populations, and so further research is warranted. However, the qualitative effects of all the studies were quite homogeneous.

## 1. Introduction

Joint range of motion (ROM) is the angle by which a joint moves from its resting position to the extremities of its motion in any given direction [1]. Improving ROM is a core goal for the general population [2], as well as in clinical contexts [3], such as in treating acute respiratory failure [4], plexiform neurofibromas [5], recovering from breast cancer-related surgery [6], and total hip replacement [7]. Several common clinical conditions negatively affect ROM, such as ankylosing spondylitis [8], cerebral palsy [9], Duchenne muscular dystrophy [10], osteoarthritis [11] rheumatoid arthritis [12]. Unsurprisingly, ROM gains are also relevant in different sports [13], such as basketball, baseball and rowing [14,15,16]. ROM is improved through increased stretch tolerance, augmented fascicle length and changes in pennation angle [17], as well as reduced tonic reflex activity [18]. Stretching is usually prescribed for increasing ROM in sports [19,20], clinical settings, such as chronic low back pain [21], rheumatoid arthritis [22], and exercise performance in general [23]. Stretching techniques, include static (active or passive), dynamic, or proprioceptive neuromuscular facilitation (PNF), all of which can improve ROM [2,24,25,26,27].

It should be noted that muscle weakness is associated with diminished ROM [28,29,30]. Strength training (ST) can be achieved through a number of methods, as long as resistance is applied to promote strength gains, and includes methods as diverse as using free weights or plyometrics [31]. Although ST primarily addresses muscle weakness, it has been shown to increase ROM [32]. For example, hip flexion and extension ROM of adolescent male hurdles was improved using plyometrics [33], while judo fighters improved ROM (shoulder flexion, extension, abduction and adduction; trunk flexion and extension; and hip flexion and extension) through resistance training [34]. The ROM gains, using resistance training, have also been described in relation to healthy elderly people for hip flexion and cervical extension [35], and isometric neck strength training, with an elastic band, in women with chronic nonspecific neck pain improved neck flexion, extension, rotation and lateral flexion [36]. ST that is focused on concentric and eccentric contractions has been shown to increase fascicle length [37,38,39]. Improvements in agonist-antagonist co-activation [40], reciprocal inhibition [41], and potentiated stretch-shortening cycles due to greater active muscle stiffness [42] may also explain why ST is a suitable method for improving ROM.

Nevertheless, studies comparing the effects of ST and stretching in ROM have presented conflicting evidence [43,44], and many have small sample sizes [45,46]. Developing a systematic review and meta-analysis may help summarize this conflicting evidence and increase statistical power, thus, providing clearer guidance for interventions [47]. Therefore, the aim of this systematic review and meta-analysis was to compare the effects of supervised and randomized ST versus stretching protocols on ROM in participants of any health and training status.

## 2. Materials and Methods

### 2.1. Protocol and Registration

The methods and protocol registration were preregistered prior to conducting the review: INPLASY, no.202090098, DOI:10.37766/inplasy2020.9.0098.

### 2.2. Eligibility Criteria

Articles were eligible for inclusion if published in peer-reviewed journals, with no restrictions in language or publication date. The Preferred Reporting Items for Systematic Reviews and Meta-Analyses (PRISMA) guidelines were adopted [48]. Participants, interventions, comparators, outcomes, and study design (P.I.C.O.S.) were established as follows: (i) Participants with no restriction regarding health, sex, age, or training status; (ii) ST interventions supervised by a certified professional. ST was defined as any method focused on developing strength, ranging from resistance training to plyometrics [31]; no limitations were placed with regard to intensity, volume, type of contractions and frequency, as it could excessively narrow the searches; (iii) comparators were supervised groups performing any form of stretching, including static stretching, passive stretching, dynamic stretching, and PNF [2], regardless of their intensity, duration or additional features; (iv) outcomes were ROM assessed in any joint, preferably through goniometry, but standardized tests such as the sit-and-reach were also acceptable; (v) randomized controlled trials (RCTs). RCTs reduce bias and better balance participant features between the groups [47], and are important for the advancement of sports science [49]. There were no limitations regarding intervention length.

The study excluded reviews, letters to editors, trial registrations, proposals for protocols, editorials, book chapters, and conference abstracts. Exclusion criteria, based on P.I.C.O.S., included: (i) Research with non-human animals; (ii) non-ST protocols or ST interventions combined with other methods (e.g., endurance); unsupervised interventions; (iii) stretching or ST + stretching interventions combined with other training methods (e.g., endurance); protocols without stretching; unsupervised interventions; (iv) studies not reporting ROM; (v) non-randomized interventions.

### 2.3. Information Sources and Search

Six databases were used to search and retrieve the articles in early October 2020: Cochrane Library, EBSCO, PubMed (including MEDLINE), Scielo, Scopus, and Web of Science (Core Collection). Boolean operators were applied to search the article title, abstract and/or keywords: (“strength training” OR “resistance training” OR “weight training” OR “plyometric*” OR “calisthenics”) AND (“flexibility” OR “stretching”) AND “range of motion” AND “random*”. The specificities of each search engine included: (i) Cochrane Library, items were limited to trials, including articles but excluding protocols, reviews, editorials and similar publications; (ii) EBSCO, the search was limited to articles in scientific, peer-reviewed journals (iii) PubMed, the search was limited to title or abstract; publications were limited to RCTs and clinical trials, excluding books and documents, meta-analyses, reviews and systematic reviews; (iv) in Scielo, Scopus and Web of Science, the publication type was limited to article; and (v) Web of Science, “topic” is the term used to refer to title, abstract and keywords.

An additional search was conducted within the reference lists of the included records. The list of articles and inclusion criteria were then sent to four experts to suggest additional references. The search strategy and consulted databases were not provided in this process to avoid biasing the experts’ searches. More detailed information is available as Appendix A.

*Updated searches:* on 8 March 2021, we conducted new searches in the databases. However, each database has specific approaches to filtering the searches by date. In Cochrane, we searched for articles entering the database in the previous 6 months. In EBSCO, we searched for all fields starting from October 2020 onwards. In PubMed, the entry date was set to 1 October 2020, onwards. In Scielo, Scopus and Web of Science, publication date was limited to 2020 and 2021.

### 2.4. Search Strategy

Here, we provide the specific example of search conducted in PubMed:


*(((“strength training” [Title/Abstract] OR “resistance training” [Title/Abstract] OR “weight training” [Title/Abstract] OR “plyometric*” [Title/Abstract] OR “calisthenics” [Title/Abstract]) AND (“flexibility” [Title/Abstract] OR “stretching” [Title/Abstract])) AND (“range of motion” [Title/Abstract])) AND (“random*” [Title/Abstract]).*


After this search, the filters RCT and Clinical Trial were applied.

### 2.5. Study Selection

J.A. and F.M.C. each conducted the initial search and selection stages independently, and then compared result to ensure accuracy. J.F. and T.R. independently reviewed the process to detect potential errors. When necessary, re-analysis was conducted until a consensus was achieved.

### 2.6. Data Collection Process

J.A., F.M.C., A.A.M. and J.F. extracted the data, while J.M., T.R., R.Z. and A.M. independently revised the process. Data for the meta-analysis were extracted by JA and independently verified by A.A.M. and R.R.C. Data were available for sharing.

### 2.7. Data Items

Data items: (i) Population: subjects, health status, sex/gender, age, training status, selection of subjects; (ii) intervention and comparators: Study length in weeks, weekly frequency of the sessions, weekly training volume in minutes, session duration in minutes, number of exercises per session, number of sets and repetitions per exercise, load (e.g., % 1 Repetition Maximum), full versus partial ROM, supervision ratio; in the comparators, modality of stretching applied was also considered; adherence rates were considered *a posteriori*; (iii) ROM testing: joints and actions, body positions (e.g., standing, supine), mode of testing (i.e., active, passive, both), pre-testing warm-up, timing (e.g., pre- and post-intervention, intermediate assessments), results considered for a given test (e.g., average of three measures), data reliability, number of testers and instructions provided during testing; (iv) Outcomes: changes in ROM for intervention and comparator groups; (vi) funding and conflicts of interest.

### 2.8. Risk of Bias in Individual Studies

The risk of bias (RoB) in individual studies was assessed using the Cochrane risk-of-bias tool for randomized trials (RoB 2) [50]. J.A. and A.M. independently completed RoB analysis, which was reviewed by F.M.C. Where inconsistencies emerged, the original articles were re-analyzed until a consensus was achieved.

### 2.9. Summary Measures

Meta-analysis was conducted when ≥3 studies were available [51]. Pre- and post-intervention means and standard deviations (SDs) for dependent variables were used after being converted to Hedges’s *g* effect size (ES) [51]. When means and SDs were not available, they were calculated from 95% confidence intervals (CIs) or standard error of mean (SEM), using Cochrane’s RevMan Calculator for Microsoft Excel [52]. When ROM data from different groups (e.g., men and women) or different joints (e.g., knee and ankle) was pooled, weighted formulas were applied [47].

### 2.10. Synthesis of Results

The inverse variance random-effects model for meta-analyses [53,54] was used to allocate a proportionate weight to trials based on the size of their individual standard errors [55], and accounting for heterogeneity across studies [56]. The ESs were presented alongside 95% CIs and interpreted using the following thresholds [57]: <0.2, trivial; 0.2–0.6, small; >0.6–1.2, moderate; >1.2–2.0, large; >2.0–4.0, very large; >4.0, extremely large. Heterogeneity was assessed using the I^2^ statistic, with values of <25%, 25–75%, and >75% considered to represent low, moderate, and high levels of heterogeneity, respectively [58]. Data used for meta-analysis is available in a Appendix A.

### 2.11. Risk of Bias Across Studies

Publication bias was explored using the extended Egger’s test [59], with *p <* 0.05 implying bias. To adjust for publication bias, a sensitivity analysis was conducted using the trim and fill method [60], with L0 as the default estimator for the number of missing studies [61].

### 2.12. Moderator Analyses

Using a random-effects model and independent computed single factor analysis, potential sources of heterogeneity likely to influence the effects of training interventions were selected, including (i) ROM type (i.e., passive versus active), (ii) studies RoB in randomization, and (iii) studies RoB in measurement of the outcome [62]. These analyses were decided post-protocol registration.

All analyses were carried out using the Comprehensive Meta-Analysis program (version 2; Biostat, Englewood, NJ, USA). Statistical significance was set at *p* ≤ 0.05. Data for the meta-analysis were extracted by JA and independently verified by A.A.M. and R.R.C.

### 2.13. Quality and Confidence in Findings

Although not planned in the registered protocol, we decided to abide by the Grading of Recommendations Assessment, Development, and Evaluation (GRADE) [63], which addresses five dimensions that can downgrade studies when assessing the quality of evidence in RCTs. RoB, inconsistency (through heterogeneity measures), and publication bias were addressed above and were considered *a priori*. Directness was guaranteed by design, as no surrogates were used for any of the pre-defined P.I.C.O. dimensions. Imprecision was assessed on the basis of 95% CIs.

## 3. Results

### 3.1. Study Selection

An initial search returned 194 results (52 in Cochrane Library, 11 in EBSCO, 11 in PubMed, 9 in Scielo, 88 in Scopus, and 23 in Web of Science). After removal of duplicates, 121 records remained. Screening the titles and abstracts for eligibility criteria resulted in the exclusion of 106 articles: 26 were not original research articles (e.g., trial registrations, reviews), 24 were out of scope, 48 did not have the required intervention or comparators, five did not assess ROM, two were non-randomized and one was unsupervised. Fifteen articles were eligible for full-text analysis. One article did not have the required intervention [64], and two did not have the needed comparators [65,66]. In one article, the ST and stretching groups performed a 20–30 min warm-up following an unspecified protocol [67]. In another, the intervention and comparator were unsupervised [68], and in one the stretching group was unsupervised [69]. Finally, in one article, 75% of the training sessions were unsupervised [70]. Therefore, eight articles were included at this stage [33,43,44,45,46,71,72,73].

A manual search within the reference lists of the included articles revealed five additional potentially fitting articles. Two lacked the intervention group required [74,75], and two were non-randomized [76,77]. One article met the inclusion criteria [78]. Four experts revised the inclusion criteria and the list of articles and suggested eight articles based on their titles and abstracts. Six were excluded: interventions were multicomponent [79,80]; comparators performed no exercise [81,82]; out of scope [83]; and unsupervised stretching group [84]. Two articles were included [85,86], increasing the list to eleven articles [33,43,44,45,46,71,72,73,78,85,86], with 452 participants eligible for meta-analysis (Figure 1). *Updated searches:* in the renewed searches, 28 records emerged, of which two passed the screening. However, these two records had already been included in our final sample. Therefore, no new article was included.

### 3.2. Study Characteristics and Results

The data items can be found in Table 1. The study of Wyon, Smith and Koutedakis [73] required consultation of a previous paper [87] to provide essential information. Samples ranged from 27 [46] to 124 subjects [43], including: Trained participants, i.e., engaging in systematic exercise programs [33,45,72,73], healthy sedentary participants [44,71,78], sedentary and trained participants [86], workers with chronic neck pain [46], participants with fibromyalgia [85], and elderly participants with difficulties in at least one of four tasks: transferring, bathing, toileting, and walking [43]. Seven articles included only women [45,73,78,85] or predominantly women [43,44,46]; three investigated only men [33,86] or predominantly men [71]; and one article had a balanced mixture of men and women [72].

Interventions lasted between five [71] and 16 weeks [78]. Minimum weekly training frequency was two sessions [46,85] and maximum was five [73]. Six articles provided insufficient information concerning session duration [44,45,72,73,78,86]. Ten articles vaguely defined training load for the ST and stretching groups [33,43,46,71,72,85,86], or for stretching groups [44,45,78]. Six articles did not report on using partial or full ROM during ST exercises [33,43,45,46,72,78]. Different stretching modalities were implemented: static active [44,46,71,78,85,86], dynamic [43,45], dynamic with a 10-s hold [33], static active in one group and static passive in another [73], and a combination of dynamic, static active, and PNF [72].

Hip joint ROM was assessed in seven articles [33,43,45,71,72,73,78], knee ROM in five [43,44,45,71,86], shoulder ROM in four [43,45,71,85], elbow and trunk ROM in two [43,45], and cervical spine [46] and the ankle joint ROM in one article [43]. In one article, active ROM (AROM) was tested for the trunk, while passive ROM (PROM) was tested for the other joints [43]. In one article, PROM was tested for goniometric assessments and AROM for hip flexion [45]. In another, AROM was assessed for the shoulder and PROM for the hip and knee [71]. Three articles only assessed PROM [44,72,86], and four AROM [33,46,78,85], while one assessed both for the same joint [73].

In seven articles [33,46,71,72,78,85], ST and stretching groups significantly improved ROM, and the differences between the groups were non-significant. In one article, the ST group had significant improvements in 8 of 10 ROM measures, while dynamic stretching did not lead to improvement in any of the groups [43]. In another article, the three groups significantly improved PROM, without between-group differences; the ST and the static active stretching groups also significantly improved AROM [73]. In two articles, none of the groups improved ROM [44,45].

### 3.3. Risk of Bias in Individual Studies

Table 2 presents assessments of RoB. Bias arising from the randomization process was low in four articles [43,45,73,85], moderate in one [46] and high in six [33,44,71,72,78,86]. Bias due to deviations from intended interventions, missing outcome data, and selection of the reported results was low. Bias in measurement of the outcome was low in six articles [44,45,46,78,85,86], but high in five [33,43,71,72,73].

### 3.4. Synthesis of Results

Comparisons were performed between ST and stretching groups, involving eleven articles and 452 participants. Global effects on ROM were achieved pooling data from the different joints. One article did not have the data required [44], but the authors supplied it upon request. For another article [45], we also requested data relative to the goniometric evaluations, but obtained no response. Therefore, only data from the sit-and-reach test were used. For one article [46], means and SDs were obtained from 95% CIs, while in another [85], SDs were extracted from SEMs using Cochrane’s RevMan Calculator.

From the five articles, including both genders, four provided pooled data, with no distinction between genders [43,44,46,71]. One article presented data separated by gender, without significant differences between men and women in response to interventions [72]. Weighted formulas were applied sequentially for combining means and SDs of groups within the same study [47]. Two studies presented the results separated by left and right lower limbs, with both showing similar responses to the interventions [33,73]; outcomes were combined using the same weighted formulas for the means and SDs. Five articles only presented one decimal place [33,43,46,72,78], and so all values were rounded for uniformity.

Effects of ST versus stretching on ROM: no significant difference was noted between ST and stretching (ES = −0.22; 95% CI = −0.55 to 0.12; *p* = 0.206; I^2^ = 65.4%; Egger’s test *p* = 0.563; Figure 2). The relative weight of each study in the analysis ranged from 6.4% to 12.7% (the size of the plotted squares in Figure 2 reflects the statistical weight of each study).

### 3.5. Additional Analyses

*Effects of ST versus stretching on ROM, moderated by study RoB in randomization*: No significant sub-group differences in ROM changes (*p =* 0.256) was found when programs with high RoB (6 studies; ES = −0.41; 95% CI = −1.02 to 0.20; within-group I^2^ = 77.5%) were compared to programs with low RoB (4 studies; ES = −0.03; 95% CI = −0.29 to 0.23; within-group I^2^ = 0.0%) (Figure 3).

*Effects of ST versus stretching on ROM, moderated by study RoB in measurement of the outcome*: No significant sub-group difference in ROM changes (*p* = 0.320) was found when programs with high RoB (5 studies; ES = −0.04; 95% CI = −0.31 to 0.24; within-group I^2^ = 8.0%) were compared to programs with low RoB (6 studies; ES = −0.37; 95% CI = −0.95 to 0.22; within-group I^2^ = 77.3%) (Figure 4).

*Effects of ST versus stretching on ROM, moderated by ROM type (active* vs. *passive)*: No significant sub-group difference in ROM changes (*p* = 0.642) was found after training programs that assessed active (8 groups; ES = −0.15; 95% CI = −0.65 to 0.36; within-group I^2^ = 78.7%) compared to passive ROM (6 groups; ES = −0.01; 95% CI = −0.27 to 0.24; within-group I^2^ = 15.3%) (Figure 5).

*Effects of ST versus stretching on hip flexion ROM:* Seven studies provided data for hip flexion ROM (pooled *n* = 294). There was no significant difference between ST and stretching interventions (ES = −0.24; 95% CI = −0.82 to 0.34; *p* = 0.414; I^2^ = 80.5%; Egger’s test *p* = 0.626; Figure 6). The relative weight of each study in the analysis ranged from 12.0% to 17.4% (the size of the plotted squares in Figure 6 reflects the statistical weight of each study).

*Effects of ST versus stretching on hip flexion ROM, moderated by study RoB in randomization:* No significant sub-group difference in hip flexion ROM changes (*p* = 0.311) was found when programs with high RoB in randomization (4 studies; ES = −0.46; 95% CI = −1.51 to 0.58; within-group I^2^ = 86.9%) were compared to programs with low RoB in randomization (3 studies; ES = 0.10; 95% CI = −0.20 to 0.40; within-group I^2^ = 0.0%) (Figure 7).

*Effects of ST versus stretching on hip flexion ROM, moderated by ROM type (active* vs. *passive):* No significant sub-group difference in hip flexion ROM changes (*p* = 0.466) was found after the programs assessed active (4 groups; ES = −0.38; 95% CI = −1.53 to 0.76; within-group I^2^ = 87.1%) compared to passive ROM (4 groups; ES = 0.08; 95% CI = −0.37 to 0.52; within-group I^2^ = 56.5%) (Figure 8).

*Effects of ST versus stretching on knee extension ROM:* Four studies provided data for knee extension ROM (pooled *n* = 223). There was no significant difference between ST and stretching interventions (ES = 0.25; 95% CI = −0.02 to 0.51; *p* = 0.066; I^2^ = 0.0%; Egger’s test *p* = 0.021; Figure 9). After the application of the trim and fill method, the adjusted values changed to ES = 0.33 (95% CI = 0.10 to 0.57), favoring ST. The relative weight of each study in the analysis ranged from 11.3% to 54.2% (the size of the plotted squares in Figure 9 reflects the statistical weight of each study).

One article behaved as an outlier in all comparisons, favoring stretching [78], but after sensitivity analysis the results remained unchanged (*p* > 0.05), with all ST versus stretching comparisons remaining non-significant.

### 3.6. Confidence in Cumulative Evidence

Table 3 presents GRADE assessments. ROM is a continuous variable, and so a high degree of heterogeneity was expected [88]. Imprecision was moderate, likely reflecting the fact that ROM is a continuous variable. Overall, both ST and stretching consistently promoted ROM gains, but no recommendation could be made favoring one protocol.

## 4. Discussion

### 4.1. Summary of Evidence

The aim of this systematic review and meta-analysis was to compare the effects of supervised and randomized ST compared to stretching protocols on ROM, in participants of any health and training status. Qualitative synthesis showed that ST and stretching interventions were not statistically different in improving ROM. However, the studies were highly heterogeneous with regard to the nature of the interventions and moderator variables, such as gender, health, or training status. This had been reported in the original manuscripts as well. A meta-analysis, including 11 articles and 452 participants, showed that ST and stretching interventions were not statistically different in active and passive ROM changes, regardless of RoB in the randomization process, or in measurement of the outcome. RoB was low for deviations from intended interventions, missing outcome data, and selection of the reported results. No publication bias was detected.

High heterogeneity is expected in continuous variables [88], such as ROM. However, more research should be conducted to afford sub-group analysis according to characteristics of the analyzed population, as well as protocol features. For example, insufficient reporting of training volume and intensity meant it was impossible to establish effective dose-response relationships, although a minimum of five weeks of intervention [71], and two weekly sessions were sufficient to improve ROM [46,85]. Studies were not always clear with regard to the intensity used in ST and stretching protocols. Assessment of stretching intensity is complex, but a practical solution may be to apply scales of perceived exertion [73], or the Stretching Intensity Scale [89]. ST intensity may also moderate effects on ROM [90], and ST with full versus partial ROM may have distinct neuromuscular effects [81] and changes in fascicle length [37]. Again, the information was insufficient to discuss these factors, which could potentially explain part of the heterogeneity of results. This precludes advancing stronger conclusions and requires further research to be implemented.

Most studies showed ROM gains in ST and stretching interventions, but in two studies, neither group showed improvements [44,45]. Although adherence rates were unreported by Aquino, Fonseca, Goncalves, Silva, Ocarino and Mancini [44], they were above 91.7% in Leite, De Souza Teixeira, Saavedra, Leite, Rhea and Simão [45], thus providing an unlikely explanation for these results. In the study by Aquino, Fonseca, Goncalves, Silva, Ocarino and Mancini [44], the participants increased their stretch tolerance, and the ST group changed the peak torque angle, despite no ROM gains. The authors acknowledged that there was high variability in measurement conditions (e.g., room temperature), which could have interfered with calculations. Leite, De Souza Teixeira, Saavedra, Leite, Rhea and Simão [45] suggested that the use of dynamic instead of static stretching could explain the lack of ROM gains in the stretching and stretching + ST groups. However, other studies using dynamic stretching have shown ROM gains [33,43]. Furthermore, Leite, De Souza Teixeira, Saavedra, Leite, Rhea and Simão [45] provided no interpretation for the lack of ROM gains in the ST group.

Globally, however, both ST and stretching were effective in improving ROM. We asked what the reason for ST to improve ROM in a manner that is not statistically distinguishable from stretching? A first thought might be to speculate that perhaps the original studies used sub-threshold stretching intensities and/or durations. However, the hypothesis that ST has intrinsic merit for improving ROM should also be considered. ST with an eccentric focus demands the muscles to produce force on elongated positions, and a meta-analysis showed limited-to-moderate evidence that eccentric ST is associated with increases in fascicle length [91]. Likewise, a recent study showed that 12 sessions of eccentric ST increased fascicle length of the biceps femoris long head [38]. However, ST with an emphasis in concentric training has been shown to increase fascicle length when full ROM was required [37]. In a study with nine older adults, ST increased fascicle length in both the eccentric and concentric groups, albeit more prominently in the former [92]. Conversely, changes in pennation angle were superior in the concentric group (35% increase versus 5% increase). Plyometric training can also increase plantar flexor tendon extensibility [42].

One article showed significant reductions in pain associated with increases in strength [46]. Therefore, decreased pain sensitivity may be another mechanism by which ST promotes ROM gains. An improved agonist-antagonist coactivation is another possible mechanism promoting ROM gains, through better adjusted force ratios [40,73]. Also, some articles included in the meta-analysis assessed other outcomes in addition to ROM, and these indicated that ST programs may have additional advantages when compared to stretching, such as greater improvements in neck flexors endurance [46], ten repetition maximum Bench Press and Leg Press [45,78], and countermovement jump and 60-m sprint with hurdles [33] which may favor the choice of ST over stretching interventions.

### 4.2. Limitations

After protocol registration, we chose to improve upon the design, namely adding two dimensions (directness and imprecision) that would provide a complete GRADE assessment. Furthermore, subgroup analyses were not planned *a priori*. There is a risk of multiple subgroup analyses generating a false statistical difference, merely to the number of analyses conducted [47]. However, all analyses showed an absence of significant differences and therefore provide a more complete understanding that the effects of ST or stretching on ROM are consistent across conditions. Looking backwards, perhaps removing the filters used in the initial searches could have provided a greater number of records. Notwithstanding, it would also likely provide a huge number of non-relevant records, including opinion papers and reviews. Moreover, consultation with four independent experts may hopefully have resolved this shortcoming.

Due to the heterogeneity of populations analysed, sub-group analysis according to sex or age group were not possible, and so it would be important to explore if these features interact with the protocols in meaningful ways. Moreover, there was a predominance of studies with women, meaning more research with men is advised. There was also a predominance of assessments of hip joint ROM, followed by knee and shoulder, with the remaining joints receiving little to no attention. In addition, dose-response relationships could not be addressed, mainly due to poor reporting. However, the qualitative findings of all the studies were very homogeneous, with statistical significance tests failing to show differences between ST and stretching protocols.

## 5. Conclusions

Overall, ST and stretching were not statistically different in ROM improvements, both in short-term interventions [71], and in longer-term protocols [78], suggesting that a combination of neural and mechanical factors is at play. However, the heterogeneity of study designs and populations precludes any definite conclusions and invites researchers to delve deeper into this phenomenon. Notwithstanding this observation, the qualitative effects were quite similar across studies. Therefore, if ROM gains are a desirable outcome, both ST and stretching reveal promising effects, but future research should better explore this avenue. In addition, the studies included in this review showed that ST had a few advantages in relation to stretching, as was explored in the discussion. Furthermore, session duration may negatively impact adherence to an exercise program [93]. If future research confirms that ST generates ROM gains similar to those obtained with stretching, clinicians may prescribe smaller, more time-effective programs when deemed convenient and appropriate, thus eventually increasing patient adherence rates. Alternatively, perhaps studies using stretching exercises should better assess their intensity and try to establish minimum thresholds for their efficacy in improving ROM.

## Figures and Tables

**Figure 1 healthcare-09-00427-f001:**
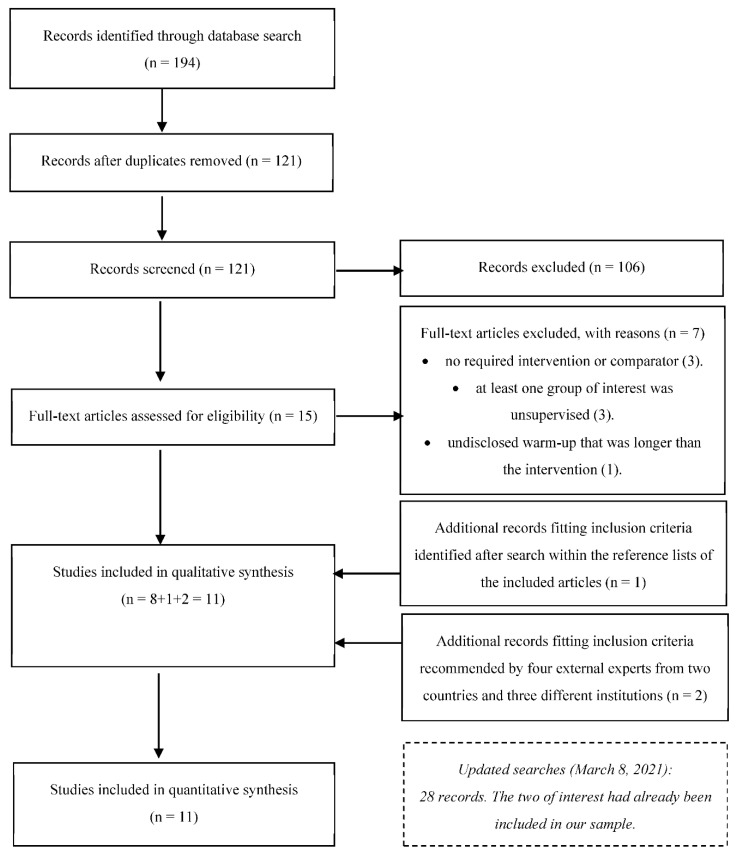
Flowchart describing the study selection process.

**Figure 2 healthcare-09-00427-f002:**
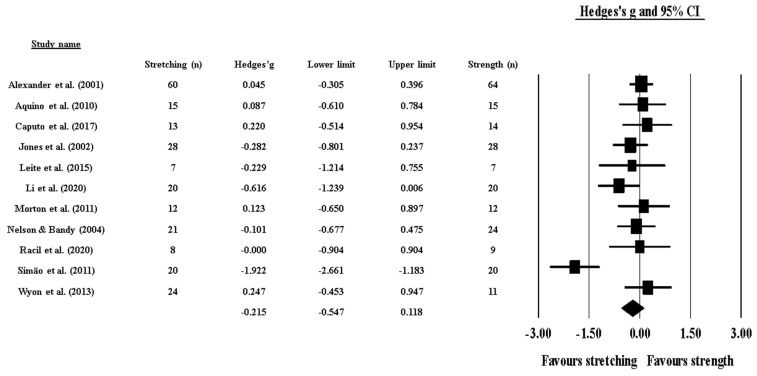
Forest plot of changes in ROM after participating in stretching-based compared to Scheme 95. confidence intervals (CI). The size of the plotted squares reflects the statistical weight of the study.

**Figure 3 healthcare-09-00427-f003:**
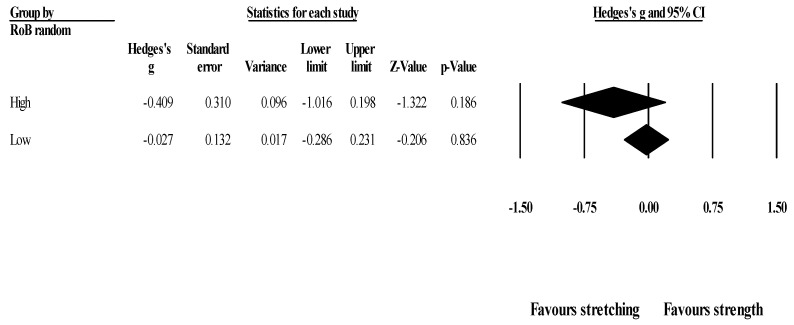
Forest plot of changes in ROM after participating in stretching-based compared to Scheme 95. confidence intervals (CI).

**Figure 4 healthcare-09-00427-f004:**
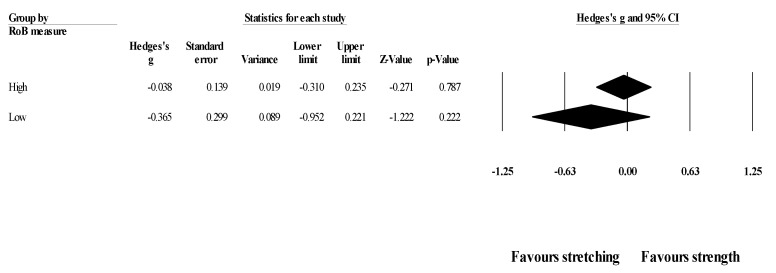
Forest plot of changes in ROM after participating in stretching-based compared to Scheme 95. confidence intervals (CI).

**Figure 5 healthcare-09-00427-f005:**
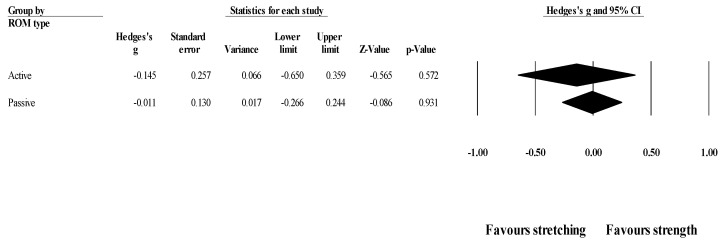
Forest plot of changes in ROM after participating in stretching-based compared to Scheme 95. confidence intervals (CI).

**Figure 6 healthcare-09-00427-f006:**
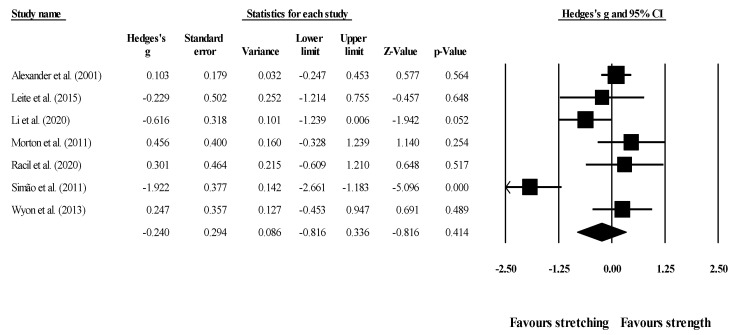
Forest plot of changes in hip flexion ROM after participating in stretching-based compared to strength-based training interventions. Values shown are effect sizes (Hedges’s g) with 95% confidence intervals (CI). The size of the plotted squares reflects the statistical weight of the study.

**Figure 7 healthcare-09-00427-f007:**
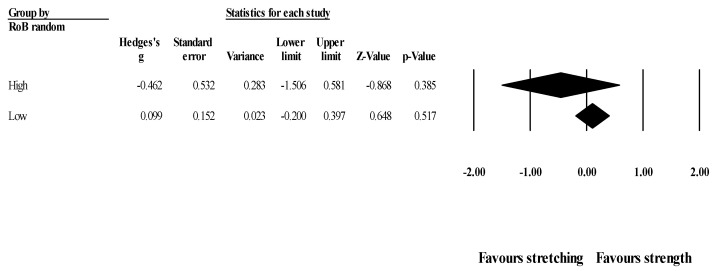
Forest plot of changes in hip flexion ROM after participating in stretching-based compared to strength-based training interventions with high versus low RoB in randomization. Values shown are effect sizes (Hedges’s g) with 95% confidence intervals (CI).

**Figure 8 healthcare-09-00427-f008:**
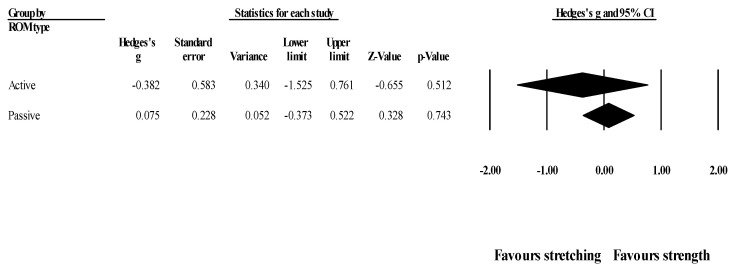
Forest plot of changes in hip flexion ROM after participating in stretching-based compared to strength-based training interventions assessing active or passive ROM. Values shown are effect sizes (Hedges’s g) with 95% confidence intervals (CI).

**Figure 9 healthcare-09-00427-f009:**
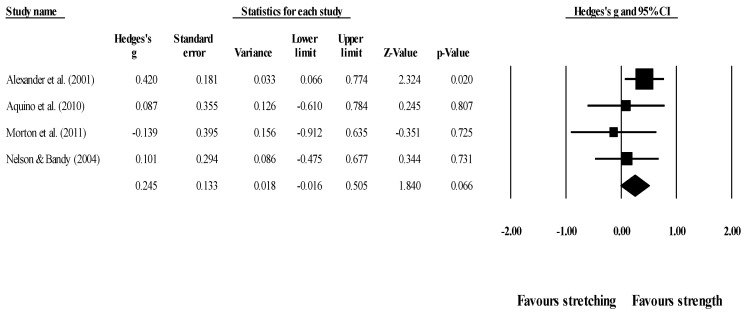
Forest plot of changes in knee extension ROM after participating in stretching-based compared to strength-based training interventions (all assessed passive ROM). Values shown are effect sizes (Hedges’s g) with 95% confidence intervals (CI). The size of the plotted squares reflects the statistical weight of the study.

**Table 1 healthcare-09-00427-t001:** Characteristics of included randomized trials.

Article	Population and Common Program Features	Strength Training Group	Comparator Group(s) *	ROM Testing	Qualitative Results for ROM
Task-Specific Resistance Training to Improve the Ability of Activities of Daily Living–Impaired Older Adults to Rise from a Bed and from a Chair. [43]	*Subjects*: 161. *Health status*: Elderly people dependent on help for performing at least one of four tasks: transferring, bathing, toileting and walking. *Gender*: ST: 84% women; STRE: 88% women. *Age*: ST: 82.0 ± 6.4; STRE: 82.4 ± 6.3. *Training status*: Not participating in regular strenuous exercise. *Selection of subjects*: Seven congregate housing facilities. *Length (weeks):* 12. *Weekly sessions*: 3. *Adherence:* average 81% of the sessions. Funding National Institute on Aging (NIA) Claude Pepper Older Adults Independence Center (Grant AG0 8808 and NIA Grant AG10542), the Department of Veterans Affairs Rehabilitation Research and Development, and the AARP-Andrus Foundation. *Conflicts of interest:* N/A.	*n* = 81 (60 completed). *Weekly volume (minutes):* 180. *Session duration in minutes:* 60. *No. exercises per session*: 16. *No. sets and repetitions*: 1*7–8 based on maximum target of 9 (bed-rise tasks). 1*5 based on maximum target of 6 (chair-rise tasks). *Load*: Unclear. Loads were incremented if subjects were not feeling challenged enough. *Full or partial ROM:* N/A. *Supervision ratio*: 1:1.	STRE *n* = 80 (64 completed). *Weekly volume (minutes):* 180. *Session duration in minutes:* 60. *No. exercises per session*: N/A. *No. sets and repetitions*: N/A. *Stretching modality*: Dynamic. *Load*: Low-intensity, but without a specified criterion. *Full or partial ROM*: N/A. *Supervision ratio*: One supervisor per group, but group size was N/A.	*Joints and actions*: Elbow (extension), shoulder (abduction), hip (flexion and abduction), knee (flexion and extension), ankle (dorsiflexion) and trunk (flexion, extension, lateral flexion). *Positions*: Supine (elbow, shoulder, hip, knee and ankle); standing (lumbar spine). *Mode*: Active for trunk, passive for the other joints. *Warm-up*: N/A. *Timing*: Baseline, 6 weeks, 12 weeks. *Results considered in the tests*: N/A. *Data reliability*: ICCs of 0.65 to 0.86 for trunk measures. Unreported for other measures. *No. testers*: N/A. *Instructions during testing*: N/A.	The ST group had significant improvements in all ROM measures, except hip flexion and abduction. The STRE group had no significant change in any of the ROM values.
Stretching versus strength training in lengthened position in subjects with tight hamstring muscles: A randomized controlled trial. [44]	*Subjects:* 45 undergraduate students. *Health status*: 30° knee extension deficit with the hip at 90° when in supine position. No injuries in the lower limbs and no lower back pain. *Gender*: 39 women, 6 men. *Age*: 21.33 ± 1.76 years (ST); 22.60 ± 1.84 years (STRE). *Training status*: No participation in ST or STRE programs in the previous year. *Selection of subjects*: announcements posted at the University. *Length (weeks):* 8. *Weekly sessions*: 3. *Adherence:* N/A. Funding N/A. *Conflicts of interest:* N/A.	*n =* 15. *Weekly volume (minutes):* N/A. *Session duration in minutes:* N/A. *No. exercises per session*: 1. *No. sets and repetitions*: 3*12. *Load*: 60% of 1RM. *Full or partial ROM*: Partial. *Supervision ratio*: N/A.	STRE *n =* 15. *Weekly volume (minutes):* N/A. *Session duration in minutes:* N/A. *No. exercises per session*: 1. *No. sets and repetitions*: 4*30″. *Stretching modality*: Static. *Load*: Unclear. *Full or partial ROM*: Full. *Supervision ratio*: N/A.	*Joints and actions*: Knee (extension). *Positions*: Sitting. *Mode*: Passive. *Warm-up*: Yes, but unclear with regard to specifications. *Timing*: Baseline, 1-week post-protocol. *Results considered in the tests*: mean of three measures. *Data reliability*: High. *No. testers*: 1 (blinded). *Instructions during testing*: N/A.	None of the groups experienced significant improvements in ROM.
Group-based exercise at workplace: short-term effects of neck and shoulder resistance training in video display unit workers with work-related chronic neck pain—a pilot randomized trial. [46]	*Subjects:* 35 video display unit workers, 27 completed the program. *Health status*: chronic neck pain. *Gender*: 27 women, 8 men. *Age*: 43 (41–45) in ST; 42 (38.5–44) in STRE. *Training status*: N/A. *Selection of subjects*: intranet form. *Length (weeks):* 7. *Weekly sessions*: 2. *Adherence:* average 85% of the sessions in ST and 86% in STRE. Funding Under “disclosures”, the authors stated “none”. *Conflicts of interest:* Under “disclosures”, the authors stated “none”.	*n =* 14. *Weekly volume (minutes):* 90. *Session duration in minutes:* 45. *No. exercises per session*: 10. *No. sets and repetitions*: 2–3*8–20. Isometric contractions up to 30″. *Load*: free-weights with a maximum of 75% MVC and elastic bands of unspecified load. *Full or partial ROM*: N/A. *Supervision ratio*: ~1:8.	STRE *n* = 13. *Weekly volume (minutes):* 90. *Session duration in minutes:* 45. *No. exercises per session*: 11. *No. sets and repetitions*: 10*10″. *Stretching modality*: Static. *Load*: N/A. *Full or partial ROM*: N/A. *Supervision ratio*: ~1:8.	*Joints and actions*: Cervical spine (flexion, extension, lateral flexion, rotation). *Positions*: Sitting (flexion, extension, lateral flexion) and supine position (rotation). *Mode*: Active. *Warm-up*: N/A. *Timing*: Baseline, 1-week post-protocol. *Results considered in the tests*: N/A. *Data reliability*: N/A. *No. testers*: 1 (blinded). *Instructions during testing*: N/A.	Significant improvements in both groups for all ROM measurements. No differences between the two groups.
A randomized controlled trial of muscle strengthening versus flexibility training in fibromyalgia. [85]	*Subjects:* 68; 56 completed the program. *Health status:* Diagnosed with fibromyalgia (FM). *Gender:* Women. *Age:* 49.2 ± 6.36 years in ST, 46.4 ± 8.56 in STRE. *Training status:* 87% were sedentary. Not engaged in regular strength training programs. *Selection of subjects:* FM patients referred to rheumatology practice at a teaching university. *Length (weeks):* 12. *Weekly sessions:* 2. *Adherence:* 85% of the initial participants attended ≥13 of 24 classes. N/A for the 46 women that completed the interventions. Funding Individual National Research Service Award (#1F31NR07337-01A1) from the National Institutes of Health, a doctoral dissertation grant (#2324938) from the Arthritis Foundation, and funds from the Oregon Fibromyalgia Foundation. *Conflicts of interest:* N/A.	*n* = 28. *Weekly volume (minutes):* 120. *Session duration in minutes:* 60. *No. exercises per session:* Presumably 12. *No. sets and repetitions:* 1*4–5, progressing to 1*12. *Load:* Low intensity. Slower concentric contractions with a 4″ isometric hold in the end, and a faster eccentric contraction. *Full or partial ROM:* Full. *Supervision ratio:* Presumably 1:28.	STRE *n* = 28. *Weekly volume (minutes):* 120. *Session duration in minutes:* 60. *No. exercises per session:* Presumably 12. *No. sets and repetitions:* N/A. *Stretching modality:* Static. *Load:* Low intensity. *Full or partial ROM:* N/A. *Supervision ratio:* Presumably 1:28.	*Joints and actions:* Shoulder (the authors report on internal and external rotation, but the movements used actually required a combination of motions). *Positions:* Presumably standing. *Mode:* Active. *Warm-up:* N/A. *Timing:* Baseline, 12 weeks. *Results considered in the tests:* N/A. *Data reliability:* Referral to a previous study, but no values for these data. *No. testers:* 1. *Instructions during testing:* Reach as far as possible.	Both groups had significant improvements in ROM. No differences between groups.
Influence of Strength and Flexibility Training, Combined or Isolated, on Strength and Flexibility Gains. [45]	*Subjects:*28 women. *Health status:* Presumably healthy. *Gender:* Women. *Age:* 46 ± 6.5. *Training status:* Trained in strength and stretching. *Selection of subjects:* Volunteers that would refrain from exercise outside the intervention. *Length (weeks):*12. *Weekly sessions:* 4. Not explicit; 48 sessions over 12 weeks. *Adherence:* Minimum was 44 of the 48 sessions. Funding N/A. *Conflicts of interest:* N/A.	*n =* 7. *Weekly volume (minutes):* N/A. *Session duration in minutes:* N/A. *No. exercises per session*: 8. *No. sets and repetitions*: 3*8–12 during the 1st month; 3*6–10RM in the 2nd month; 3*10–15RM in the 3rd month. *Load*: 6–15RM, depending on the month. *Full or partial ROM*: N/A. *Supervision ratio*: N/A.	STRE *n* = 7. *Weekly volume (minutes):* 240. *Session duration in minutes:* 60. *No. exercises per session*: N/A. *No. sets and repetitions*: 3*30. *Stretching modality*: Dynamic. *Load*: Stretch to mild discomfort. *Full or partial ROM*: Full. *Supervision ratio*: N/A. STRE + ST *n* = 7. Completion of both protocols. Unknown duration. ST + STRE *n* = 7. Completion of both protocols in reverse order. Unknown duration.	*Joints and actions:* Shoulder (flexion, extension, abduction and horizontal adduction) elbow (flexion), hip (flexion and extension), knee (flexion), and trunk (flexion and extension). *Positions:* Supine (shoulder flexion, abduction, horizontal adduction, elbow and hip flexion), prone (shoulder and hip extension, and knee flexion) and upright (trunk flexion and extension) for goniometric evaluations. Sitting for sit-and-reach. *Mode:* Passive for goniometry. Active for sit-and-reach. *Warm-up:* 5- minute walking on treadmill at mild to moderate intensity and four stretching exercises. *Timing:* Baseline, 12 weeks. *Results considered in the tests:* Best of 3 trials. *Data reliability:* Very high. *No. testers:* 1. *Instructions during testing:* N/A.	None of the groups experienced significant improvements in ROM.
Effects of Flexibility and Strength Interventions on Optimal Lengths of Hamstring Muscle-Tendon Units. [72]	*Subjects:* 40 college students. *Health status*: No history of lower extremity injury in the 2 years prior to the study. *Gender*: 20 men, 20 women. *Age*: 18–24 years. *Training status*: participating in exercise 2–3 times per week. *Selection of subjects*: college students. *Length (weeks):* 8. *Weekly sessions*: 3. *Adherence:* N/A. Funding Partially supported by the National Natural Science Foundation of China (Grant No.: 81572212) and the Fundamental Research Fund for the Central Universities, Beijing Sport University (Grant No.: 2017XS017). *Conflicts of interest:* N/A.	*n* = 20. *Weekly volume (minutes):* N/A. *Session duration in minutes:* N/A *No. exercises per session*: 4. *No. sets and repetitions*: 2–4*8–15. For one exercise, 2*50–60″. *Load*: N/A. *Full or partial ROM*: N/A. *Supervision ratio*: N/A.	STRE *n* = 20. *Weekly volume (minutes):* N/A. *Session duration in minutes:* N/A. *No. exercises per session*: 4. *No. sets and repetitions*: 2*15 for dynamic stretching, 2*40–60″ for static stretching, 3*50″ for PNF and 3*40–50″ for foam roll. *Stretching modality*: active static, dynamic and PNF. *Load*: N/A. *Full or partial ROM*: N/A. *Supervision ratio*: N/A.	*Joints and actions*: Hip (flexion). *Positions*: Supine. *Mode*: Passive. *Warm-up*: Six-minute warm-up including jogging and jumping. *Timing*: Baseline, 8 weeks. *Results considered in the tests*: Mean of three trials. *Data reliability*: Very high. *No. testers*: N/A. *Instructions during testing*: N/A.	Men and women in both groups significantly improved ROM. No differences between groups.
Resistance training vs. static stretching: Effects on flexibility and strength. [71]	*Subjects:* 37 college students. *Health status*: Healthy. *Gender*: 30 men, 12 women; ratio is unclear in the final sample. *Age*: 21.91 ± 3.64 years. *Training status*: Untrained. *Selection of subjects*: Recruited from Physical Education or Exercise Science Classes. *Length (weeks):* 5. *Weekly sessions*: 3. *Adherence:* N/A. Funding N/A. *Conflicts of interest:* N/A.	*n =* 12. *Weekly volume (minutes):* 135–180. *Session duration in minutes:* 45–60. *No. exercises per session*: eight in days 1 and 2, four in day 3. *No. sets and repetitions*: 4 sets of unspecified repetitions. *Load*: N/A. *Full or partial ROM*: Full. *Supervision ratio*: 1:12.	STRE *n =* 12. *Weekly volume (minutes):* 75–90. *Session duration in minutes:* 25–35. *No. exercises per session*: 13. *No. sets and repetitions*: 1*30″ for most stretches. 3*30″ for one exercise and 3*20″ for two. *Stretching modality*: Static. *Load*: N/A. *Full or partial ROM*: Full. *Supervision ratio*: 1:12.	*Joints and actions*: Hip (flexion, extension), knee (extension) and shoulder (extension). *Positions*: Supine (knee and hip). Prone (shoulder). *Mode*: Passive for hip and knee, active for shoulder. *Warm-up*: 5 min of stationary bicycle with minimal resistance. *Timing*: Baseline, 1-week post-protocol. *Results considered in the tests*: N/A. *Data reliability*: N/A. *No. testers*: 1. *Instructions during testing*: Technical instructions specific to each test.	Both groups had significant improvements in knee extension, hip flexion and hip extension, but not shoulder extension. No differences between the interventions.
Eccentric training and static stretching improve hamstring flexibility of high school men. [86]	*Subjects:* 69 high-schoolers. *Health status*: Healthy, but with a 30° loss of knee extension. *Gender*: Men. *Age*: 16.45 ± 0.96 years. *Training status:* Some sedentary, others involved in exercise programs. *Selection of subjects*: Volunteers with tight hamstrings. *Length (weeks):* 6. *Weekly sessions*: 3 for STRE. N/A for ST. *Adherence:* N/A for ST. STRE: subjects missing >4 sessions were excluded. Funding N/A. *Conflicts of interest:* N/A.	*n =* 24. *Weekly volume (minutes):* N/A. *Session duration in minutes:* N/A. *No. exercises per session*: 1. *No. sets and repetitions*: 6 repetitions with 5″ isometric hold between each. *Load*: N/A. *Full or partial ROM*: Full. *Supervision ratio*: Description suggests a 1:1 ratio.	STRE *n=* 21. *Weekly volume (minutes):* N/A. *Session duration in minutes:* N/A. *No. exercises per session*: 1. *No. sets and repetitions*: Unknown number of repetitions, each lasting 30″. *Stretching modality*: Static. *Load*: Stretch until a gentle stretch was felt on the posterior thigh. *Full or partial ROM*: Full. *Supervision ratio*: Description suggests a 1:1 ratio.	*Joints and actions*: Knee (extension). *Positions*: Supine. *Mode*: Passive. *Warm-up*: No warm-up. *Timing*: Baseline, 6 weeks. *Results considered in the tests*: Two measures for previous reliability calculations, but unclear for the groups’ evaluation. *Data reliability*: Very high. *No. testers*: 2 (1 blinded). *Instructions during testing*: N/A.	Both groups improved ROM. No differences between the interventions.
Effects of flexibility combined with plyometric exercises vs. isolated plyometric or flexibility mode in adolescent men hurdlers. [33]	*Subjects:* 34 trained hurdlers. *Health status*: No lower extremity injury in the previous 30 days. *Gender*: Men. *Age*: 15 ± 0.7 years. *Training status*: ≥3 years of experience in hurdle racing. *Selection of subjects*: Recruited from three athletic teams. *Length (weeks):* 12. *Weekly sessions*: 4. *Adherence:* N/A. Funding No funding. *Conflicts of interest:* No conflicts of interest.	*n = 9*. *Weekly volume (minutes):* 320. *Session duration in minutes:* 80. *No. exercises per session*: 4. *No. sets and repetitions*: 3*30″. *Load*: Evolved from low to hard intensity, but no criteria were provided. *Full or partial ROM*: N/A. *Supervision ratio*: N/A.	STRE *n = 8*. *Weekly volume (minutes):* 320. *Session duration in minutes: 80*. *No. exercises per session*: 7. *No. sets and repetitions*: 5*10″. *Stretching modality*: Dynamic with 10″ static hold. *Load*: Evolved from low to hard intensity, but no criteria were provided. *Full or partial ROM*: N/A. *Supervision ratio*: N/A. STRE+ ST *n = 9*. *Weekly volume (minutes):* 320. *Session duration in minutes:* 80. *No. exercises per session*: 11 (4 plyometric; 7 flexibility). *No. sets and repetitions*: 3*30″ for plyometrics, 5*10″ for stretching. *Stretching modality*: Dynamic with 10″ static hold. *Load*: Evolved from low to hard intensity, but no criteria were provided. *Full or partial ROM*: N/A. *Supervision ratio*: N/A.	*Joints and actions*: Hip (flexion and extension). *Positions*: Supine. *Mode*: Active. *Warm-up*: N/A. *Timing*: Baseline, 12 weeks. *Results considered in the tests*: Best of 3 attempts. *Data reliability*: Referral to a previous study, but no values for these data. *No. testers*: 2. *Instructions during testing*: N/A.	All interventions had significant improvements in ROM. No differences between the interventions.
The influence of strength, flexibility, and simultaneous training on flexibility and strength gains. [78]	*Subjects:* 80 women. *Health status:* Healthy. *Gender:* Women. *Age:* 35 ± 2.0 (ST), 34 ± 1.2 (STRE), 35 ± 1.8 (ST + STRE), 34 ± 2.1 (non-exercise). *Training status:* Sedentary. *Selection of subjects:* Volunteers that were sedentary ≥12 months. *Length (weeks):* 16. *Weekly sessions:* 3. *Adherence:* Minimum was 46 of the 48 sessions. Funding N/A. *Conflicts of interest:* N/A.	*n* = 20. *Weekly volume (minutes):* N/A. *Session duration in minutes:* N/A. *No. exercises per session:* 8. *No. sets and repetitions:* 3*8–12 in the 1st and 4th months; 3*6–10 in 2nd month; 3*10–15 in 3rd month. *Load:* 8–12RM (1st and 4th months); 6–10RM (2nd month); 10–15RM (3rd month). *Full or partial ROM:* N/A. *Supervision ratio:* N/A.	STRE *n* = 20. *Weekly volume (minutes):* N/A. *Session duration in minutes:* N/A. *No. exercises per session:* N/A. *No. sets and repetitions:* 4*15–60″. Duration of each set started at 15″ and progressed to 60″ during the intervention. *Stretching modality:* Static. *Load:* Performed at the point of mild discomfort. *Full or partial ROM:* N/A. *Supervision ratio:* N/A. ST + STRE *n* = 20. STRE protocol followed by the ST protocol.	*Joints and actions:* Hip (flexion) and knee (extension) combined. *Positions:* Sitting. *Mode:* Active. *Warm-up:* 4 stretching exercises (2*10″). *Timing:* Baseline, 16 weeks. *Results considered in the tests:* Maximum of 3 attempts. *Data reliability:* Very high. *No. testers:* 1. *Instructions during testing:* N/A.	The interventions significantly improved ROM. No differences between the interventions.
A comparison of strength and stretch interventions on active and passive ranges of movement in dancers: a randomized control trial. [73]	*Subjects:* 39 dance students, 35 completed. *Health status*: N/A. *Gender*: Women (39). *Age*: 17 ± 0.49 years (ST group); 17 ± 0.56 years (low-intensity STRE); 17 ± 0.56 years (moderate to high intensity STRE). *Training status*: Moderately trained dance students. *Selection of subjects*: Recruited from dance college. *Length (weeks):* 6. *Weekly sessions*: 5. *Adherence:* N/A. Funding N/A. *Conflicts of interest:* N/A.	*n =* 11. *Weekly volume (minutes):* N/A. *Session duration in minutes:* N/A. *No. exercises per session*: 1. *No. sets and repetitions*: 3*5, increasing to 3*10 during the program. Each repetition included a 3″ isometric hold. *Load*: Unclear, but using body weight. *Full or partial ROM*: Partial (final 10°). *Supervision ratio*: N/A.	Low-intensity STRE *n =* 13. *Weekly volume (minutes):* N/A. *Session duration in minutes:* N/A. *No. exercises per session*: 5. *No. sets and repetitions*: N/A, but 1′ for each stretch. *Stretching modality*: Active static. *Load*: 3/10 perceived exertion. *Full or partial ROM*: N/A. *Supervision ratio*: N/A. Moderate-intensity or high-intensity STRE *n =* 11. *Weekly volume (minutes):* N/A. *Session duration in minutes:* N/A. *No. exercises per session*: 5. *No. sets and repetitions*: N/A. *Stretching modality*: Passive. *Load*: 8/10 perceived exertion. *Full or partial ROM*: N/A. *Supervision ratio*: N/A.	*Joints and actions*: Hip (flexion). *Positions*: Standing. *Mode*: Active and passive. *Warm-up*: 10 min of cardiovascular exercise and lower limb stretches. *Timing*: Baseline, 6 weeks. *Results considered in the tests*: N/A. *Data reliability*: N/A. *No. testers*: N/A. *Instructions during testing*: Positioning cues for ensuring proper posture.	The three groups significantly improved passive ROM, without differences between the groups. The moderate-to-high intensity STRE group did not improve in active ROM. The two other interventions did.

Legend: N/A—Information not available. ST—Strength training. STRE—Stretching. ROM—Range of motion. MVC—Maximum voluntary contraction. PNF—Proprioceptive neuromuscular facilitation. * Non-exercise groups are not considered in this column.

**Table 2 healthcare-09-00427-t002:** Assessments of risk of bias (Cochrane’s RoB 2).

Article	Randomization Process	Deviations from Intended Interventions (EAI ^1^)	Missing Outcome Data	Measurement of the Outcome	Selection of the Reported Results
Alexander, Galecki, Grenier, Nyquist, Hofmeyer, Grunawalt, Medell and Fry-Welch [43]				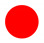	
Aquino, Fonseca, Goncalves, Silva, Ocarino and Mancini [44]	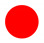				
Caputo, Di Bari and Naranjo Orellana [46]	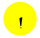				
Jones, Burckhardt, Clark, Bennett and Potempa [85]					
Leite, De Souza Teixeira, Saavedra, Leite, Rhea and Simão [45]					
Li, Garrett, Best, Li, Wan, Liu and Yu [72]	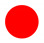			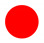	
Morton, Whitehead, Brinkert and Caine [71]	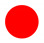			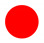	
Nelson and Bandy [86]	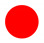				
Racil, Jlid, Bouzid, Sioud, Khalifa, Amri, Gaied and Coquart [33]	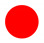			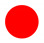	
Simão, Lemos, Salles, Leite, Oliveira, Rhea and Reis [78]	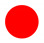				
Wyon, Smith and Koutedakis [73]				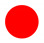	

^1^ Effect of assignment to intervention. 

 Low risk of bias; 
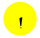
 Some concerns; 
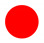
 High risk of bias.

**Table 3 healthcare-09-00427-t003:** GRADE assessments for the certainty of evidence.

Outcome	Study Design	RoB ^1^	Publication Bias	Inconsistency	Indirectness	Imprecision	Quality of Evidence	Recommendation
ROM	11 RCTs, 452 participants in meta-analysis.	*Randomization*—low in four articles, moderate in one, and high in six.	No publication bias.	9 RCTs showed improvements in ROM in both groups. 2 RCTs showed no changes in ROM in either group. 11 RCTs showed effects of equal magnitude for ST and stretching. ^2^	No serious indirectness.	Moderate. ^3^	Moderate. ⨁⨁⨁	Moderate recommendation for either strength training or stretching ^4^. No recommendation for choosing one of the protocols over the other, as their efficacy in ROM gains was statistically not different.
		*Deviations from intended interventions*—Low.						
		*Missing outcome data*—Low.						
		*Measurement of the outcome*—low in six articles and high in five.						
		*Selection of the reported results*—Low.						

1—Meta-analyses moderated by RoB showed no differences between studies with low and high risk. 2—Because ROM is a continuous variable, high heterogeneity was expected. However, this heterogeneity is mostly between small and large beneficial effects. No adverse effects were reported. 3—Expected because ROM is a continuous variable. Furthermore, imprecision referred to small to large beneficial effects. 4—Both strength training and stretching presented benefits without reported adverse effects.

## Data Availability

Our data were made available with the submission.

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
