# Peer review of "Strength Training versus Stretching for Improving Range of Motion: A Systematic Review and Meta-Analysis"

_healthcare, 2021, doi:10.3390/healthcare9040427_

Round 1

Reviewer 1 Report

The article aims meta-analyze randomized controlled trials assessing the effects of ST and stretching on ROM. The study included 11 articles (n = 37 452 participants). Pooled data showed no differences between ST and stretching on 38 ROM (ES = -0.22; 95% CI = -0.55 to 0.12; p = 0.206). The authors concluded that ST and stretching were not different in their effects on ROM. The article is interesting and well written. I would like to see its publication if the authors consider my suggestions.

First, the authors should describe which common diseases are manifested by limitation of joint range of motions. This is to highlight the clinical importance of limitation in joint range of motion.

Second, please reference the following two articles in the portion describing the random effect model: Utility of sonoelastography for the evaluation of rotator cuff tendon and pertinent disorders: a systematic review and meta-analysis. Eur Radiol. 2020; The Prevalence of Sarcopenia and Its Impact on Clinical Outcomes in Lumbar Degenerative Spine Disease-A Systematic Review and Meta-Analysis. J Clin Med. 2021.

Third, in Figure 1, please edit the squares to let them present in a similar size. Please let the connecting lines and arrows either in the horizontal or vertical directions.

Fourth, please add some clarifications regarding the signs and colors in Table 2.  

Fifth, in Figure 2, the study names, effect sizes and case numbers are not in the same line. By the way, where are the 95% confidence intervals? Please redraw the figure.

Author Response

We thank the reviewer for the very kind assessment of our work. With regard to the reviewer’s suggestions:

  • First suggestion: we now describe several common diseases that are manifested by limitation of range of motion, therefore highlighting the clinical importance of limitation in joint range of motion. This is placed in the first paragraph and identified with Track Changes.
  • Second suggestion: these two papers are now referenced. Because Track Changes was not assuming the new EndNote fields (as there was no new text), this part is highlighted in red color.
  • Third suggestion: figure 1 was re-edited and we hope this new version is better.
  • Fourth suggestion: clarifications added to table 2 in the form of a legend.
  • Fifth suggestion: figure 2 was redrawn.

Reviewer 2 Report

Dear Authors,

Congratulations on your work. The objective behind your work is remarkable and has called my attention before in a work by the same some authors (https://doi.org/10.6063/motricidade.20191). With this SR I think the "towards a new paradigm" has begun. Nevertheless, particularly I consider that the results fell in the opposite direction of your expectations. Fortunately, one can perform SR and check. Above all, under all studies limitations, some insight arose for future research. Congrats on that too.

I made my revision using a checklist that I have uploaded. I did not find any major issues in your SR. All the required points are present and have a supporting rationale.

I am not willing to describe all the check-list as I could not fin any fatal flaw but I would like to highlight that statistics are well described and adequate to the analysis and sample, as well as the Risk of Bias. The authors commented on the limitations of the review itself, including those of the included studies (well done).

The only thing I have found that probably is wrong is reference 36 in Table 2. In the column of "measurements of the outcome", reference 36 should be in red. I have checked the supplementary files. 

My revision files will be uploaded along with the chek-list (I usually do that), but my only concern is about table 2.

Congratulations on your work. It is not normal to revise an almost flawless work. I will keep following your research with even more interest.

( There is no need for authors to read the PRISMA checklist and handwriting annotations. Please check table 2 (RoB).)

Author Response

We thank the reviewer for the very kind assessment of our work. With regard to the reviewer’s suggestions:

  • Table 2: it was indeed a mistake, and we have now corrected. We also double checked with our original full-assessment tables. Furthermore, table 2 now has a proper legend.

However, the reference numbers have now changed. At the request of reviewer 1, we added extra citations in the introduction and in the methods, and so former reference #36 is now #43.

Round 2

Reviewer 1 Report

The article is fantastically revised. Well done!